



# SO$_2$ enhances aerosol formation from anthropogenic volatile organic compound ozonolysis by producing sulfur-containing compounds

Zhaomin Yang[1], Kun Li[1], Narcisse T. Tsona[1], Xin Luo[2], Lin Du[1]

[1]Environment Research Institute, Shandong University, Qingdao, 266237, China

[2]Technology Center of Qingdao Customs, Qingdao, 266003, China

*Correspondence to*: Lin Du (lindu@sdu.edu.cn)

**Abstract.** Sulfur dioxide (SO$_2$) can affect aerosol formation in the atmosphere, but the

underlying mechanisms remain unclear. Here, we investigate aerosol formation and composition from the ozonolysis of cyclooctene with and without SO$_2$ addition in a smog chamber. Liquid chromatography equipped with high-resolution tandem mass spectrometry measurements indicate that monomer carboxylic acids and corresponding dimers with acid anhydride and aldol structures are important components in particles

formed in the absence of SO$_2$. A 9.4–12.6 time increase in particle maximum number concentration is observed in the presence of 14–192 ppb SO$_2$. This increase is largely attributed to sulfuric acid (H$_2$SO$_4$) formation from the reactions of stabilized Criegee intermediates with SO$_2$. In addition, a number of organosulfates (OSs) are detected in the presence of SO$_2$, which are likely products formed from the heterogeneous reactions

of oxygenated species with H$_2$SO$_4$. The molecular structures of OSs are also identified based on tandem mass spectrometry analysis. It should be noted that some of these OSs have been found in previous field studies but were classified as compounds from unknown sources or of unknown structures. The observed OSs are less volatile than their precursors and therefore are more effective contributors to particle formation and

growth, partially leading to the increase in particle volume concentration under SO$_2$-presence conditions. Our results provide an in-depth molecular-level insight into how SO$_2$ alters particle formation and composition.

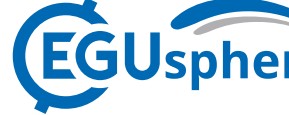

## 1 Introduction

Secondary organic aerosol (SOA) accounts for a large fraction of the organic
aerosol mass. The atmospheric oxidation of anthropogenic volatile organic compounds
(AVOCs) can produce low-volatility organic products that contribute to SOA formation
and growth (Kelly et al., 2018; Fan et al., 2020). The oxidation of AVOCs can dominate
SOA formation under severe haze episodes (Nie et al., 2022; He et al., 2020; Huang et
al., 2019; Qiu et al., 2020). Thus, AVOCs have been commonly considered as
significant SOA precursors. SOA can negatively impact air quality, global climate, and
public health (Nault et al., 2021; Zhu et al., 2017). To better understand air pollution
and develop effective particle control strategies, it is necessary to investigate the
formation mechanism and molecular composition of anthropogenic SOA.

Recently, the impacts of inorganic gases on anthropogenic SOA formation have
received significant attention. In particular, there is increasingly much evidence that
sulfur dioxide ($SO_2$) can modulate SOA formation and composition (Ye et al., 2018;
Stangl et al., 2019; Liu et al., 2017). Liu et al. (2017) reported that SOA formation from
cyclohexene photooxidation was inhibited by atmospherically relevant concentrations
of $SO_2$, as a result of the reaction of hydroxyl radical ($\cdot OH$) with $SO_2$ (to form sulfuric
acid ($H_2SO_4$)) competing with the $\cdot OH$ reaction with cyclohexene. They demonstrated
that $H_2SO_4$-catalyzed SOA enhancement was not sufficient to compensate for the loss
of $\cdot OH$ reactivity towards cyclohexene, leading to the suppression in cyclohexene SOA
formation. On the other hand, $SO_2$ can enhance SOA formation by interacting with
organic peroxide or stabilized Criegee intermediate (sCI) during the ozonolysis of
alkenes (Stangl et al., 2019; Ye et al., 2018). Organosulfates (OSs) were observed in the
above studies and have also been detected in different $SO_2$-alkene interaction areas
(Hettiyadura et al., 2019; Wang et al., 2018; Bruggemann et al., 2020). OSs are
ubiquitous in atmospheric aerosols and may be used as tracers of SOA influenced by
$SO_2$ emissions (Bruggemann et al., 2020). To further gain mechanistic insights into the



complex roles of $SO_2$ in SOA formation, it is important to explore the chemical nature and formation mechanism of OSs.

Cycloalkenes emitted from diesel vehicles and industrial processes are a crucial class of AVOCs in the atmosphere. They can be used to explore key chemical processes involved in atmospheric oxidation and SOA formation (Räty et al., 2021). However,

SOA formation chemistry from cycloalkenes has received less attention than that from linear or branched alkenes, leading to significant uncertainties in our understanding of SOA. Recent studies have reported that ozonolysis of cycloalkenes could form highly oxidized products and have considerable SOA yield (Räty et al., 2021; Rissanen, 2018). Among the most common cycloalkenes (with 5 to 8 carbon atoms), cyclooctene has the

largest potential to SOA formation (Keywood et al., 2004). Ozonolysis is the dominant oxidation pathway of cyclooctene with a reaction rate constant of $4.51 \times 10^{-16}$ $cm^3$ $molecule^{-1}$ $s^{-1}$ (298 K). Urban atmosphere is highly complex and may contain various concentrations of cycloalkenes and $SO_2$, which complicates SOA formation and composition. While most previous studies have identified compounds containing

carbon, hydrogen, and oxygen atoms (CHO compounds) as important contributors to cycloalkene SOA (Hamilton et al., 2006; Gao et al., 2004; Räty et al., 2021), the potential of OS formation from the ozonolysis of cyclooctene in the presence of $SO_2$ and the chemical processes behind OS formation remain unclear.

Given the significance of cycloalkene and $SO_2$ emissions in aerosol formation, we

investigated the $SO_2$ effects on the formation and chemical composition of cyclooctene SOA. Aerosol particles were formed from the ozonolysis of cyclooctene in the absence and presence of $SO_2$ in a smog chamber. Structural identifications of the observed products were reported and corresponding formation mechanisms were proposed. We report the mechanism showing how $SO_2$ impacts particle formation and growth based

on the observation of sulfuring-containing compounds. Our results provide a more comprehensively mechanistic understanding of the roles of $SO_2$ in modulating SOA formation and composition.



## 2 Experimental methods

### 2.1 Particle production

85        Particle formation from the ozonolysis of cyclooctene was carried out under dark conditions in a 1.2 $m^3$ Teflon chamber housed in a temperature-controlled room. A summary of experimental conditions and results is listed in Table 1. Detailed experimental equipment and methods have been described in our previous studies (Yang et al., 2022; Yang et al., 2021). Briefly, cyclooctene was introduced into the

chamber by passing zero air through a tube containing a known volume of cyclooctene (95%, Alfa). Then, cyclohexane (99.5%, Aladdin) was injected into the chamber to scavenge more than 95% of the ·OH generated during cyclooctene ozonolysis. When desired, $SO_2$ was added to the chamber from a $SO_2$ calibration cylinder. Initial concentration ratios of $SO_2$ to cyclooctene were in the range of ~0.07–1 ppb $ppb^{-1}$ to

simulate different polluted atmospheric conditions. The reactor was stabilized for 20 min under dark conditions to allow for mixing of species. Finally, ozonolysis of cyclooctene was initiated by introducing $O_3$ produced via a commercial ozone generator (WH-H-Y5Y, Wohuan, China). All experiments were performed at room temperature (~295 K) and atmospheric pressure (~1 atm) without seed particles. Temperature and

relative humidity (RH) inside the chamber were measured with a hygrometer (Model 645, Testo AG, Germany). $O_3$ and $SO_2$ concentrations over the course of ozonolysis were monitored by a Thermo Scientific model 49i $O_3$ analyzer and a Thermo Scientific model 43i-TLE $SO_2$ analyzer, respectively. The detection limits of $O_3$ analyzer and $SO_2$ analyzer were 0.5 ppb and 0.05 ppb, respectively. Size distributions and volume

concentrations of particles were continuously recorded using a scanning mobility particle sizer (SMPS), which consisted of differential mobility analyzer (Model 3082, TSI, USA) and ultrafine condensation particle counter (Model 3776, TSI, USA).





**Table 1.** Experimental conditions and results for particle formation experiments.

| [Cyclooctene]$_0$ (ppb) | [O$_3$]$_0$ (ppb) | T (K) | RH (%) | SO$_2$ (ppb) | $\triangle$SO$_2$ [a] (ppb) | $V_{H2SO4}$ [b] ($\mu m^3\ cm^{-3}$) | $N_{max}$ [c] $\times10^6$ ($cm^{-3}$) | $V_{particle}$ [d] ($\mu m^3\ cm^{-3}$) |
|---|---|---|---|---|---|---|---|---|
| 195 | 839 | 296 | 25 | - | - | - | 0.14 | 151 ± 2.9 |
| 195 | 770 | 294 | 24 | 14 | 3.7 | 9.4 | 1.31 | 170 ± 5.7 |
| 195 | 800 | 294 | 22 | 28 | 8.9 | 22.6 | 1.05 | 191 ± 5.5 |
| 195 | 792 | 293 | 21 | 50 | 9.6 | 24.5 | 1.06 | 228 ± 4.2 |
| 195 | 730 | 292 | 20 | 100 | 9.5 | 24.3 | 1.54 | 264 ± 7.4 |
| 195 | 790 | 293 | 23 | 154 | 17.2 | 43.8 | 1.28 | 270 ± 5.0 |
| 191 | 743 | 295 | 24 | 192 | 14.2 | 35.9 | 1.77 | 280 ± 8.2 |

[a] $\triangle$SO$_2$ represents the consumed SO$_2$ concentration during the ozonolysis of cyclooctene.

[b] The volume concentration of particle-phase H$_2$SO$_4$ assuming a full conversion of SO$_2$ to H$_2$SO$_4$ with a density of 1.58 g cm$^{-3}$ under moderate humidity conditions (Wyche et al., 2009; Ye et al., 2018).

[c] $N_{max}$ denotes the maximum number concentration of aerosol particles during the ozonolysis of cyclooctene.

[d] $V_{particle}$ is the volume concentration of aerosol particles, which has been corrected for wall loss of particles. Errors represent standard deviation for particle formation experiments.

## 2.2 Particle collection and chemical characterization

Aerosol particles were collected on aluminum foils using a 14-stage low-pressure impactor (DLPI+, Dekati Ltd, Finland). All samples were stored in -20 ℃ freezer until analysis. Offline functional group measurements of aerosol particles were performed using an attenuated total reflectance-Fourier transform infrared spectrometer (ATR-FTIR, Vertex 70, Bruker, Germany). Before each measurement, the diamond crystal was thoroughly cleaned with ethanol and ultrapure water to eliminate the interference of ambient contaminants on functional group measurements of aerosol particles. ATR-FTIR spectra of blank aluminum foils and aerosol samples were recorded in the range



of 4000–600 cm$^{-1}$ at a resolution of 4 cm$^{-1}$ with 64 scans. The data of ATR-FTIR spectra were recorded with the OPUS software.

Aerosol particles were also collected on polytetrafluoroethylene (PTFE) filters (0.22 μm pore size, 47 mm dimeter, TJMF50, Jinteng, China). Sample filters were

extracted twice into 5 mL of methanol (Optima® LC-MS grade, Fisher Scientific) by ice sonication for 20 min. Extracts were then filtered, concentrated to near dryness and subsequently reconstituted in 200 μL of 50:50 (v/v) methanol and ultrapure water. Blank filters were also subjected to the same extraction and preparation procedure. Obtained extracts of blank and sample filters were analyzed using a Thermo Scientific

ultrahigh-performance liquid chromatograph, which was coupled with a high-resolution Q Exactive Focus Hybrid Quadrupole-Orbitrap mass spectrometer equipped with an electrospray ionization (ESI) source (UHPLC/ESI-HRMS). Samples were first separated on an Atlantis T3 C18 column (100 Å pores, 3 μm particle size, 2.1 mm × 150 mm, Waters, USA) at 35 ℃. The used binary mobile phase system consisted of

ultrapure water with 0.1% (v/v) formic acid (A) and methanol with 0.1% (v/v) formic acid (B). The LC gradient employed was as follows: 0–3 min at 3% B, 3–25 min increased linearly to 50% B, 25–43 min ramped linearly to 90% B, 43−48 min returned to 3% B, and 48−60 min B held constant at 3% to re-equilibrate the column. The injected volume of samples and flow rate were 2 μL and 200 μL min$^{-1}$, respectively.

The ESI source was operated in both positive (+) and negative (−) ion modes to ionize analyte components with a scan range of mass-to-charge ($m/z$) 50 to 750. LC/ESI-MS parameter settings were as follows: 3.5 kV spray voltage (+), −3.0 kV spray voltage (−), 50 V S-lens radio frequency (RF) level (+), 50 V S-lens RF level (−), 320 ℃ capillary temperature, $2.76 × 10^5$ Pa sheath gas (nitrogen) pressure, and 3.33 L min$^{-1}$ auxiliary

gas (nitrogen) flow. Data-dependent tandem mass spectrometry (MS/MS) analysis were also carried out by high-energy collision-induced dissociation (CID) with stepped collision energies of 20, 40, and 60 eV. For MS/MS experiments, an isolation width of 2 $m/z$ units was applied. Other parameters were also selected in MS/MS experiments as





follows: $2 \times 10^5$ automatic gain control (AGC) target, 50 ms maximum IT, 3 loop count,

$1 \times 10^5$ minimum AGC target, 2–6 s apex trigger, and 6 s dynamic exclusion. The mass

resolution of MS and MS/MS were 70000 (full width at half maximum, FWHM, at $m/z$

200) and 17500, respectively. Detailed data processes are reported elsewhere (Yang et

al., 2021; Yang et al., 2022).

The double bond equivalent (DBE) value is the combined number of rings and

double bonds in the product $C_cH_hO_oN_nS_s$ and could be calculated according to eq. 1.

$$\text{DBE} = 1 + c + \frac{n - h}{2} \tag{1}$$

Kendrick mass defect (KMD) analysis could provide chemical insights into

chemical compositions of complex organic mixtures. $CH_2$ and the oxygen atom (O) are

usually chosen as base units for Kendrick analysis of complex organic mass spectra.

Kendrick mass (KM) could be converted into a new mass scale from the IUPAC mass

(eq. 2 and 4). KMD is determined as the difference between the nominal mass of a

compound (the rounded integer mass) and KM (eq. 3 and 5).

$$\text{KM}_{CH_2} = m/z \times \frac{14.00000}{14.01565} \tag{2}$$

$$\text{KMD}_{CH_2} = \text{Nominal mass} - \text{KM}_{CH_2} \tag{3}$$

$$\text{KM}_O = m/z \times \frac{16.00000}{15.99492} \tag{4}$$

$$\text{KMD}_O = \text{Nominal mass} - \text{KM}_O \tag{5}$$

The saturation mass concentration ($C^o$, μg m$^{-3}$) of product $i$ was also calculated

based on its elemental composition using the following expression (Li et al., 2016):

$$\log_{10}C_i^o = (n_C^0 - n_C^i)b_C - n_O^i b_O - 2\frac{n_C^i n_O^i}{n_C^i + n_O^i}b_{CO} - n_S^i b_S \tag{6}$$

where $n_C^0$ is the reference carbon number; $n_C^i$, $n_O^i$, and $n_S^i$ represent the numbers of

carbon, oxygen, and sulfur atoms, respectively; $b_C$, $b_O$, and $b_S$ denote the

contribution of each carbon, oxygen, and sulfur atom to $\log_{10}C_i^o$; and $b_{CO}$ is the

carbon–oxygen nonideality.





### 2.3 Wall loss corrections

The wall loss rates of $O_3$ and $SO_2$ inside the chamber were determined to be $2.05 \times 10^{-4}$ min$^{-1}$ and $2.02 \times 10^{-4}$ min$^{-1}$ (Fig. S1), respectively, indicating that the losses of these two gas-phase species to the chamber walls were negligible over the course of experiments. The wall loss of cyclooctene ($5.23 \times 10^{-6}$ min$^{-1}$) was also negligible while its oxidation products may deposit to the inner walls. However, wall losses of gas-phase products could be mitigated due to excess $O_3$ concentration. The quick oxidation and

nucleation could provide attractive condensation surfaces for oxidation products, thereby reducing the product wall losses to some extent (Stirnweis et al., 2017). Although wall losses of organic vapors may underestimate the particle mass, this work mainly focuses on the characterization of particle composition rather than the absolute SOA yield.

Independent wall-loss experiments of ammonium sulfate (($NH_4$)$_2$SO$_4$) particles were also performed to determine the size-dependent wall-loss rate constants of particles inside the chamber. An aqueous ($NH_4$)$_2$SO$_4$ solution was added to a TSI Model 3076 atomizer to produce droplets. The droplets were passed through a silica gel diffusion dryer to get dry ($NH_4$)$_2$SO$_4$ particles and then were injected into the chamber.

The size distributions of ($NH_4$)$_2$SO$_4$ particles were characterized using the SMPS for 6 h. The relationship between the wall-loss rate ($k$, h$^{-1}$) of particles and their size ($d_p$, nm) can be expressed as $k(d_p) = 1.20 \times 10^{-7} \times d_p^{2.32} + 20.59 \times d_p^{-1.39}$ based on size-dependent particle wall-loss correction method.

### 3 Results and discussion

### 3.1 SO$_2$ effects on aerosol formation

Insights into $SO_2$ effects on particle formation could be gained through investigating the number and volume concentration as well as size distribution of particles under various $SO_2$ level conditions. In the absence of $SO_2$, the particle number



concentration increased burst within the first 20 min of cyclooctene ozonolysis and then

decreased because of their coagulation and wall depositions, while the particle volume

concentration increased gradually and reached its maximum within 240 min (Fig. S2).

Elevating $SO_2$ level can result in significant increases in the number and volume

concentration of particles (Fig. 1a), which is consistent with observations from previous

studies (Ye et al., 2018; Yang et al., 2021). We observed a 9.4–12.6 time increase in

particle maximum number concentration in the presence of 14–192 ppb $SO_2$ (Table 1).

The promoted effect of $SO_2$ is shown more clearly in Fig. 1b, where $SO_2$ was seen to

be consumed on similar timescale as particle formation. Specifically, upon initiation of

cyclooctene ozonolysis, $SO_2$ concentration decreased and the particle volume

concentration increased simultaneously. After cyclooctene was completely consumed,

both $SO_2$ consumption and particle production slowed down. $SO_2$ consumption and

particle formation resumed when more cyclooctene was introduced into the reactor.

This result indicates that $SO_2$ may react with certain highly reactive species produced

from cyclooctene ozonolysis. For instance, reactions of $SO_2$ with sCI could form $H_2SO_4$

(Boy et al., 2013), which is a key specie for new particle formation (Lehtipalo et al.,

2018; Yao et al., 2018). Inorganic sulfate absorption at $617 \ cm^{-1}$ was observed in the

ATR-FTIR spectra of particles formed in cyclooctene/$O_3$/$SO_2$ systems (Fig. 2)

(Hawkins et al., 2010; Coury and Dillner, 2008), indicating the formation of $H_2SO_4$. We

assumed that all consumed $SO_2$ was converted to particle-phase $H_2SO_4$, which

represents an upper limit of the $H_2SO_4$ formation (Wyche et al., 2009; Ye et al., 2018).

The amount of $H_2SO_4$ produced could not fully account for the enhancement of particle

volume concentration (Table 1). $H_2SO_4$ has been considered as an important driver of

particle acidity (Tilgner et al., 2021). Acid catalysis induced by $H_2SO_4$ may also

promote the formation of additional organic products, leading to the increase in particle

volume concentration (Deng et al., 2021).

SO_2 can also affect the growth of new aerosol particles (Fig. 1c). In the initial stage

of ozonolysis (10 and 60 min), particles formed in cyclooctene/$O_3$/$SO_2$ systems had a





smaller size mode than those formed in cyclooctene/$O_3$ system, which may be attributed to the following two factors. First, oligomers formed from sCI reactions with organic species could partition into the condensed phase to contribute to particle growth (Riva et al., 2017). $SO_2$ presence may lead to the decrease in these oligomers because $SO_2$ can compete with organic species to react with sCI. Second, counterbalancing the reduction of oligomers via sCI + $SO_2$ reactions is the production of $H_2SO_4$. The production of more new particles in cyclooctene/$O_3$/$SO_2$ systems could provide more condensation sinks. Organic vapors that can condense onto particles are dispersed via new particles, resulting in small particle size at the initial phase of cyclooctene/$O_3$/$SO_2$ systems (Stangl et al., 2019). Interestingly, particles could grow quickly in the presence of $SO_2$. At 300 min reaction time, particles formed in the presence of $SO_2$ even had slightly larger sizes than those formed in the absence of $SO_2$. $H_2SO_4$-catalyzed heterogenous reactions could produce lower-volatile organic species from higher-volatile reactants in the aerosol phase (Yang et al., 2020; Han et al., 2016). Semi-volatile species could undergo evaporation after partitioning to the aerosol phase while low-volatile products generally have a negligible evaporation rate from the aerosol phase. Low-volatile products formed via $H_2SO_4$-catalyzed heterogenous reactions could build particle mass at a rate almost equal to the condensation rate and thus effectively facilitate the particle growth in cyclooctene/$O_3$/$SO_2$ systems (Apsokardu and Johnston, 2018).





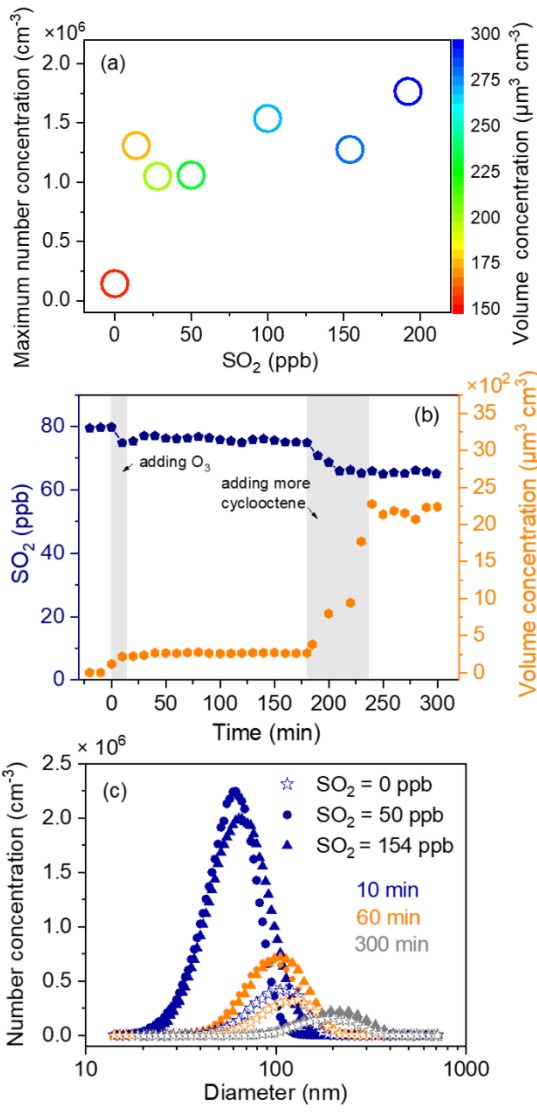

**Figure 1.** Particle formation from the ozonolysis of cyclooctene under various $SO_2$
conditions. (a) Maximum particle number concentration as a function of initial $SO_2$
level. Circle color represents particle volume concentration. (b) Temporal profiles of
$SO_2$ concentration and particle volume concentration. (c) Size distributions of aerosol
particles formed with various $SO_2$ concentrations at 10, 60, and 300 min after the after
the initiation of cyclooctene ozonolysis.



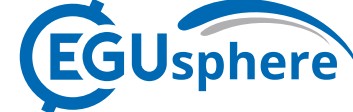

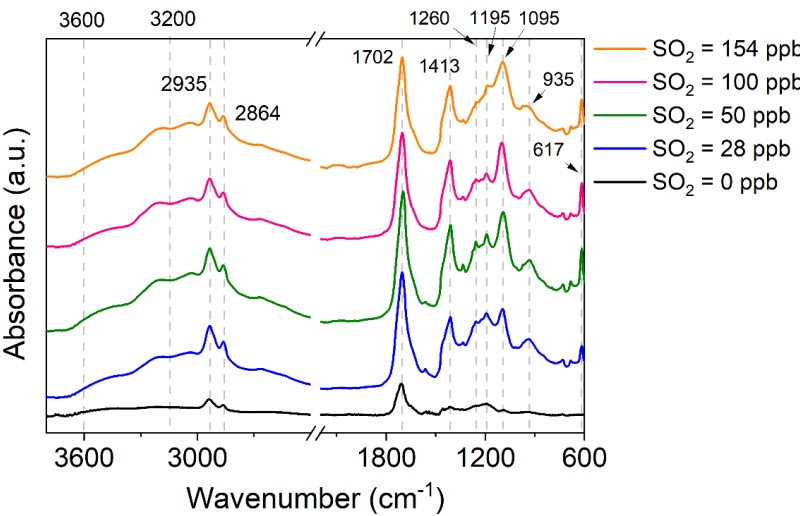

**Figure 2.** ATR-FTIR spectra of aerosol particles generated from cyclooctene ozonolysis in the presence of different $SO_2$ concentration.

### 3.2 Aerosol chemical composition under $SO_2$-free condition

Figure 3 shows the base peak chromatograms (BPCs) of cyclooctene-derived particles in the absence of $SO_2$. The chromatograms of blank filter showed clearly no peaks eluted at retention times (RTs) between 0 and 30 min while there were several significant peaks for cyclooctene SOA chromatograms in both positive and negative ion modes. Each chromatogram peak of cyclooctene SOA represents at least one ion,

and major peaks are only labeled with the mass of the most abundant single ion. Compared to the negative chromatogram of cyclooctene SOA, the corresponding label ions in the positive chromatogram were 24 Da higher in mass. This is consistent with the fact that many ions produce adducts with sodium ion ($[M + Na]^+$) in positive ion mode, while negative ion mode leads to the production of deprotonated ions ($[M – H]^-$)

(Mackenzie-Rae et al., 2018). From Fig. 3, products with molecular weight (MW) < 200 Da eluted from the column at shorter RT than those with MW > 200 Da. Low-molecular-weight products (MW < 200 Da) likely correspond to small monomer type compounds (hereafter termed as monomeric products), which are directly originated



from the ozonolysis of cyclooctene. Compounds with MW > 200 Da mainly dominate the later part of the chromatogram, and they may be homo or heterodimeric species (hereafter noted as dimeric products) formed using two monomeric products as building blocks.

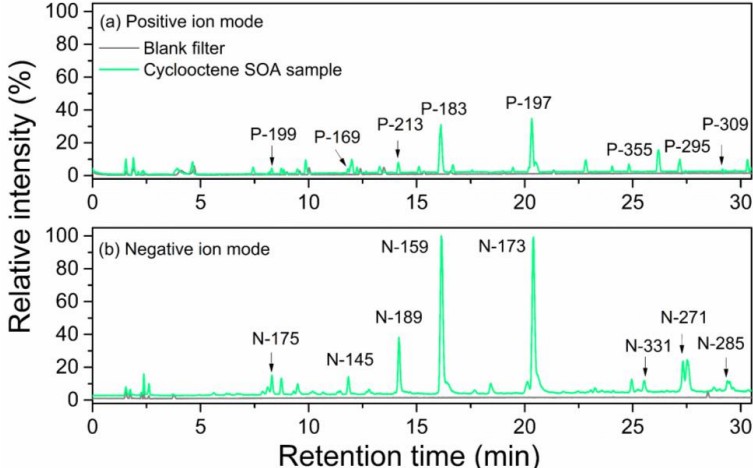

**Figure 3.** Base peak chromatograms of both blank filter and particles generated from the ozonolysis of cyclooctene in the absence of $SO_2$. Labels represent the most abundant single ion of each peak. (a) Positive ion mode. (b) Negative ion mode.

Possible structures of major monomeric products were proposed based on their accurate *m/z*, fragmentation mass spectra, and previous mechanistic insights. Note that the fragmentation of $[M + Na]^+$ is relatively difficult (Zhao et al., 2016) and, thus, the positive ion mode was not further analyzed in providing structural insights in the current study. The negative chromatogram peaks with RT at 11.85 min (N-145), 16.13 min (N-159), and 20.41 min (N-173) were significant peaks for cyclooctene SOA (Fig. 3b), and they were assigned neutral formulas of $C_6H_{10}O_4$, $C_7H_{12}O_4$, and $C_8H_{14}O_4$, respectively. As shown in Fig. 4, MS/MS spectra of monomer $C_6H_{10}O_4$, $C_7H_{12}O_4$, and $C_8H_{14}O_4$ were similar. Taking $C_8H_{14}O_4$ as example (Fig. 4c), its fragmentation mass spectrum was characterized by a loss of 44 Da ($CO_2$), suggesting the presence of carboxyl group. The neutral loss of 18 Da ($H_2O$) upon fragmentation of the parent ion ($C_8H_{13}O_4^-$, *m/z* =



173.08209) led to the production of an ion with $m/z$ 155.07143. The loss of $H_2O$ is an

unspecific fragmentation mechanism, which is likely originated from a carboxyl or

hydroxyl group (Noziere et al., 2015). The fragment ion ($m/z = 111.08166$) representing

the simultaneous neutral losses of $CO_2$ and $H_2O$ was also formed. MS/MS spectra can

be resulted from multiple isomeric structures in many cases (Wang et al., 2019).

Yasmeen et al. (2011) showed the detailed fragmentation spectrum for the dicarboxylic

acid standard (azelaic acid) and indicated that deprotonated azelaic acid also showed

losses of $H_2O$, $CO_2$, and $CO_2 + H_2O$. In addition, Noziere et al. (2015) showed that the

neutral losses of $CO_2$ and $H_2O$ indicates two carboxyl groups. Thus, monomer $C_8H_{14}O_4$

was tentatively assigned to suberic acid and the corresponding fragmentation pathways

for $C_8H_{13}O_4^-$ is proposed in Fig. S3. The fragment ions originated from losses of $H_2O$,

$CO_2$, and $CO_2 + H_2O$ were also observed in MS/MS spectra of $C_6H_{10}O_4$ and $C_7H_{12}O_4$,

indicative of adipic acid and pimelic acid, respectively. Carboxylic acids have also been

observed in SOA produced from previous alkene ozonolysis (Hamilton et al., 2006;

Kenseth et al., 2020; Mackenzie-Rae et al., 2018; Zhang et al., 2015). Carboxylic acids

represent a significant class of aerosol components, and they play a significant role in

particle chemistry by their influences on particle acidity and through direct involvement

in certain heterogeneous reactions to produce low volatile species (Millet et al., 2015).

More experiments using available authentic standards are necessary to better

understand their structures, sources, and formation mechanism. Other prominent

monomer peaks at RTs 8.30 min (N-175) and 14.18 min (N-189) corresponded to

compounds with neutral formula, namely $C_7H_{12}O_5$ and $C_8H_{14}O_5$. The losses of $H_2O$,

CO, and $CO_2$ in MS/MS spectrum of $C_7H_{12}O_5$ indicated hydroxyl, terminal carbonyl,

and carboxyl group, respectively (Mackenzie-Rae et al., 2018; Riva et al., 2016a), and

$C_7H_{12}O_5$ was identified as hydroxy-containing oxoheptanoic acid (Fig. S4a and S4c).

Monomer $C_8H_{14}O_5$ only showed losses of $H_2O$ and CO (Fig. S4b), and it is difficult to

determine the specific type and positioning of oxygen-containing functionalities within

$C_8H_{14}O_5$ with 5 oxygen atoms based on its MS/MS spectrum.




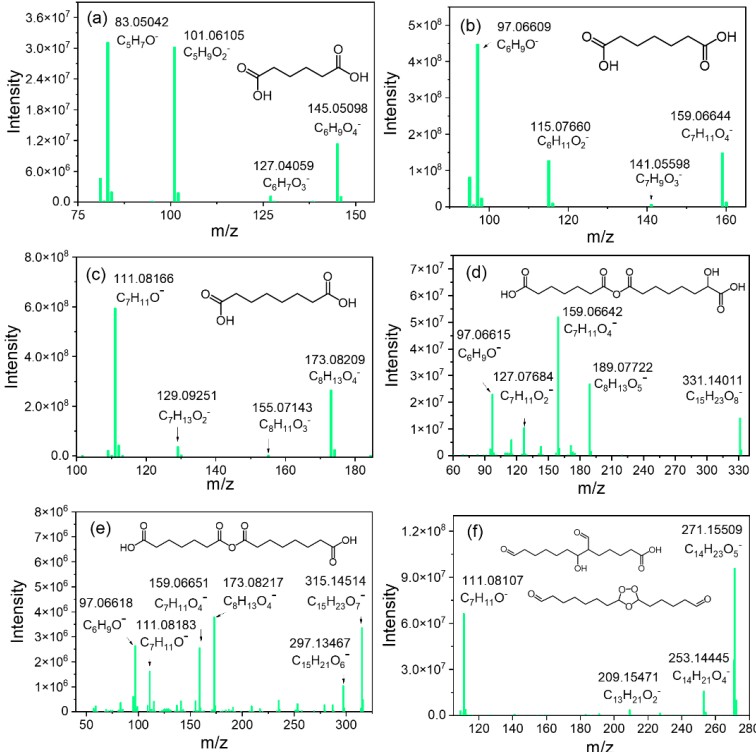

**Figure 4.** MS/MS spectra of major monomers and dimers. Monomers: (a) $C_6H_{10}O_4$, (b) $C_7H_{12}O_4$, and (c) $C_8H_{14}O_4$. Dimers: (d) $C_{15}H_{24}O_8$, (e) $C_{15}H_{24}O_7$, and (f) $C_{14}H_{24}O_5$.

The labeled dimer peaks in negative ion mode corresponded to $[M - H]^-$ ion masses of 271, 285, and 331 (Fig. 3b), which were assigned neutral formulas of $C_{14}H_{24}O_5$, $C_{15}H_{26}O_5$, and $C_{15}H_{24}O_8$, respectively. The number of fragment ions of dimers are generally limited, and determining the exact structure of dimers is less certain compared to monomers (Witkowski and Gierczak, 2017). Therefore, only a decrease in molecular

structure possibilities is provided. For dimer $C_{15}H_{24}O_8$, fragment ions *m/z* 159.06642 ($C_7H_{11}O_4^-$) and *m/z* 189.07722 ($C_8H_{13}O_5^-$) were detected in its MS/MS spectrum (Fig. 4d). When dimers are subjected to CID, fragment ions corresponding to their building blocks are commonly observed (Witkowski and Gierczak, 2017; Hall and Johnston, 2012). Based on this rule, it could be concluded that dimer $C_{15}H_{24}O_8$ was an association

product of $C_7H_{12}O_4$ and $C_8H_{14}O_5$. Similarly, for dimer $C_{15}H_{24}O_7$, there were two





significant product ions of $C_{15}H_{23}O_7^-$ with accurate masses of *m/z* 159.06651 ($C_7H_{11}O_4^-$) and 173.08217 ($C_8H_{13}O_4^-$) (Fig. 4e). Furthermore, fragment ions corresponding to secondary loss of $CO_2 + H_2O$ from product ions $C_7H_{11}O_4^-$ and $C_8H_{13}O_4^-$ were also observed. The fragmentation spectrum of $C_{15}H_{24}O_7$ was similar to the MS/MS spectra

of $C_7H_{12}O_4$ and $C_8H_{14}O_4$ (Fig. 4b and 4c), suggesting again that $C_7H_{12}O_4$ and $C_8H_{14}O_4$ may be the building blocks of $C_{15}H_{24}O_7$. Acid-catalyzed heterogeneous processes can result in the formation of high-molecular-weight dimers in both biogenic and anthropogenic systems (Barsanti et al., 2017). Carboxylic acid monomers formed could be important sources of particle acidity in the absence of $SO_2$. Dimers $C_{15}H_{24}O_7$ and

$C_{15}H_{24}O_8$ may be produced by heterogeneous reactions involving the loss of a water molecule, and the linkage between building blocks is an acid anhydride (Fig. S5) (Hamilton et al., 2006). Another abundant dimer peak (N-271) in negative chromatogram was identified as $C_{14}H_{23}O_5^-$ with mass accuracy of -0.02492 ppm. $C_{14}H_{23}O_5^-$ could dissociate to the product ions of $C_{14}H_{21}O_4^-$, $C_{13}H_{21}O_2^-$, and $C_7H_{11}O^-$.

(Fig. 4f). Both secondary ozonide and aldol structures shown in Fig. 4f could match the assigned elemental formula of $C_{14}H_{24}O_5$. However, the neutral losses of $H_2O$ and $CO_2$ were not easily produced by secondary ozonide, but more likely for the aldol structure (Hall and Johnston, 2012). Aldol condensation products were also one of the most commonly observed species in previous ozonolysis of alkenes (Zhao et al., 2016;

Kenseth et al., 2018; Kristensen et al., 2016). Therefore, $C_{14}H_{24}O_5$ shown in Fig. 4f is likely an aldol condensation product.

To examine the overall composition of particles, average mass spectra (Fig. S6) corresponding to the chromatogram where particle components eluted were also analyzed. Figure 5 summarizes the oxidation products observed in particles mapped in

O-KMD and van Krevelen plot. The molecular formulas of identified oxidation products could be largely classified into homologous series of monomers and dimers (Fig. 5a and 5b). The elemental composition distribution of products measured in positive and negative ion modes was similar, with most monomers and dimers having



O/C ratios ranging from 0.2 to 0.8, and H/C ratios ranging from 1.2 to 1.8 (Fig. 5c).

Lines with slopes of 0, -0.5, -1, and -2 in Fig. 5c can be used to illustrate the addition of hydroxyl/peroxide, carboxylic acid (with fragmentation), carboxylic acid (without fragmentation), and carbonyl groups to a saturated carbon chain, respectively (Heald et al., 2010). As shown in Fig. 5c, cyclooctene SOA occupied a relatively wide range in the van Krevelen diagram, and the large number of points scattered in the space between

lines with slopes of -0.5 and -2. This behavior is consistent with the importance of high abundance carboxylic acids in the above analysis.

### 3.3 SO$_2$ effects on aerosol chemical composition

To further get detailed mechanisms about SO$_2$ effects and determine whether heterogeneous processes occurred, aerosol samples were analyzed using ATR-FTIR and

LC/ESI-MS. Both IR and MS analysis of particles revealed changes in aerosol chemical composition induced by SO$_2$ addition.

### 3.3.1 Characteristics of functional group in aerosol-phase products

Figure 2 shows ATR-FTIR spectra of aerosol particles. Hydroxy (3600–3200 cm$^{-1}$), alkyl (2935 and 2864 cm$^{-1}$), and carbonyl (1702 cm$^{-1}$) were identified in particles

collected from the cyclooctene/O$_3$ system. These particles also had a broad absorption across the 1500–800 cm$^{-1}$ region, which may arise from C–H deformation in 1480–1350 cm$^{-1}$, C–C stretching in 1250–1120 cm$^{-1}$, and C–O stretching in different regions for various oxygenated species (Hung et al., 2013). For particles formed in the presence of SO$_2$, new absorption bands at 1413, 1095, and 617 cm$^{-1}$ were observed, which could

be ascribed to the asymmetric SO$_2$ stretching of RO−S(O)$_2$−OR' in organosulfates, symmetric SO$_2$ stretching of organic or inorganic sulfates, and the absorption of inorganic sulfates (Hawkins et al., 2010; Coury and Dillner, 2008), respectively. These remarkably different IR absorption features suggest that the SO$_2$ addition can lead to the formation of sulfur-containing compounds.





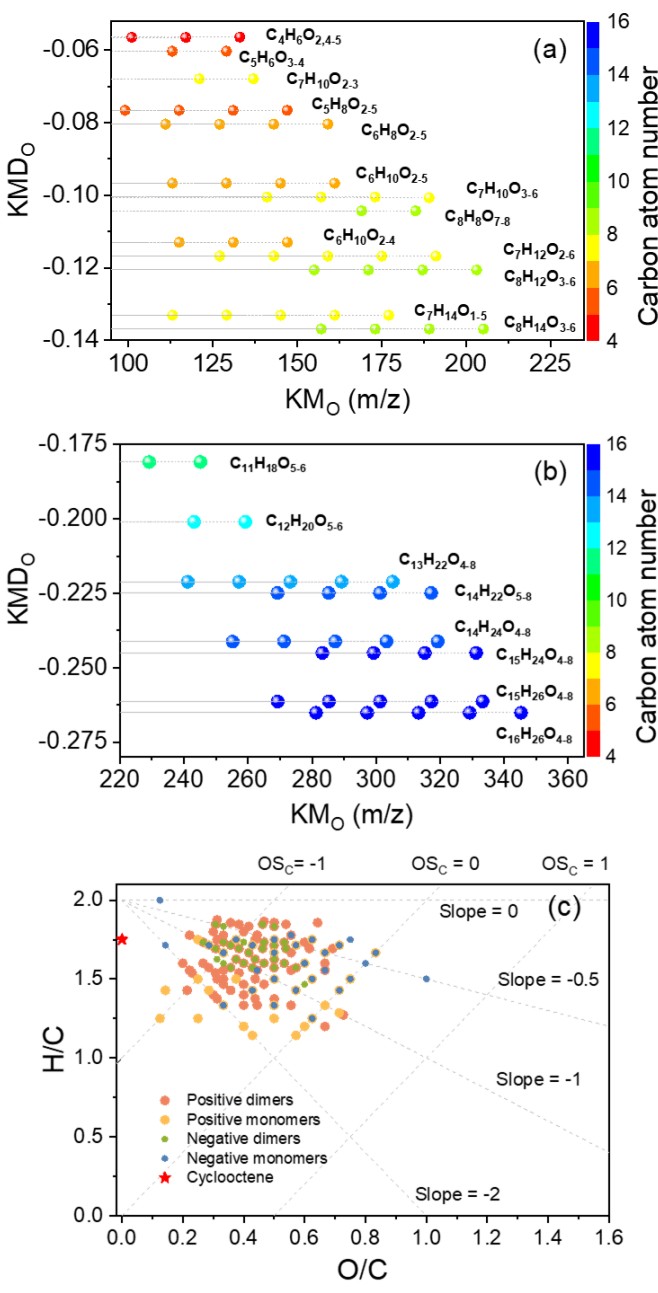

**Figure 5.** Oxidation products observed in particles produced from the ozonolysis of cyclooctene in the absence of $SO_2$. Oxygen (O)-Kendrick mass defect plots of (a) monomers and (b) dimers. (c) Van Krevelen diagram.





### 3.3.2 Organosulfate formation in the presence of SO₂

In addition to CHO compounds, products with $C_cH_hO_oS_s$ elemental formulas were identified in the presence of $SO_2$ (Fig. S7), further supporting the production of organosulfur species. Most of the observed organosulfur compounds were identified as OSs based on their accurate masses and the appearance of characteristic product ions at $m/z$ 80 ($SO_3^-$), 81 ($HSO_3^-$), and/or 97 ($HSO_4^-$) in their MS/MS spectra (Figs. S8–S16). Accurate mass measurements of OSs as well as their retention times and DBE values are provided in Table S1. The proposed structure and fragmentation scheme of each OS and corresponding precursor are presented in Figs. S8–S16. For instance, OS-209 and OS-223 showed prominent product ions for losses of $HSO_4^-$ and $SO_3^-$ (Figs. S10–S11), confirming the organosulfate moiety. Neither a hydroxyl nor a carboxyl group fragment ion (i.e., $-H_2O$ or $-CO_2$) was observed in their MS/MS spectra. $C_6H_{10}O_3$ and $C_7H_{12}O_3$ were proposed as the precursor of OS-209 and OS-223, respectively. MS/MS spectra of $C_6H_{10}O_3$ and $C_7H_{12}O_3$ were characterized by loss of CO, indicating terminal carbonyl group (Figs. S10–S11). Considering structural features of OS precursor measurements as well as OS-209 and OS-223 all corresponding to DBE = 2, two terminal carbonyl groups could explain well the observed MS/MS spectra of OS-209 and OS-223. The organosulfate substituent was expected to attach to internal carbon atom. Although the carbonyl group is more readily observed in positive ion mode, ESI-MS is also highly sensitive to carbonyl compounds containing sulfate substituents and thereby gives intense $[M - H]^-$ ions in negative ion mode (Riva et al., 2016b).

Relatively high abundance of OS is helpful for the acquisition of MS/MS data, and therefore high abundance $[M - H]^-$ ions were chosen as representative candidates to clarify the precursors and formation pathways of OSs. Simplified chemical mechanism describing OS production from the ozonolysis of cyclooctene ($C_8H_{14}$) is proposed in Fig. 6. The ozonolysis of cyclooctene ($C_8H_{14}$) can be initiated by $O_3$ addition to the endocyclic double bond, forming an energy-rich primary ozonide (POZ). POZ can decompose rapidly to an excited CI containing both a terminal carbonyl and carbonyl



oxide group. The excited CI could lead to the formation of sCI, vinylhydroperoxide, and dioxirane, illustrating the multiplicity and the complexity of cyclooctene ozonolysis.

SCI is mainly capable of involving in bimolecular reactions to form carboxylic acids and acid esters. Vinylhydroperoxide rapidly decomposes into an alkyl radical ($C_8H_{13}O_2\cdot$) and an $\cdot OH$. Molecular oxygen could be subsequently added to $C_8H_{13}O_2\cdot$ to produce an alkyl peroxy radical (RO$_2$, $C_8H_{13}O_4\cdot$). Dioxirane intermediate may also undergo decomposition and produce a $C_7H_{13}O_3\cdot$. $C_8H_{13}O_4\cdot$ and $C_7H_{13}O_3\cdot$ are considered as the

starting point of the RO$_2\cdot$ and alkoxy radical (RO$\cdot$) chemistry, resulting in termination CHO compounds with hydroperoxy, carbonyl, or hydroxy groups (Fig. 6). Acid-catalyzed heterogenous reactions of CHO products have been evidenced to play a major role in OS formation in the atmosphere (Riva et al., 2016c; Riva et al., 2016b). Although acidic seed particles were not directly injected into the reactor during cyclooctene

ozonolysis, SO$_2$-induced $H_2SO_4$ may create acidic conditions for the occurrence of heterogeneous reaction. In the case of CHO products with hydroxyl group, $H_2SO_4$ could protonate the hydroxyl group, leading to the formation of OS and water. The low RH (~ 20%) of ozonolysis was helpful for shifting the reaction equilibrium in favor of OS production.

Detailed information about the volatility of oxidation products is necessary to evaluate their potential to contribute to aerosol formation. As shown in Fig. 7a, the products could be categorized into intermediate volatility OCs (IVOCs), semi-volatile OCs (SVOCs), low-volatile OCs (LVOCs), and extremely low-volatile OCs (ELVOC) with $C^o$ in the range of 300–3 × 10$^6$, 0.3–300, 3 × 10$^{-4}$–0.3, and < 3 × 10$^{-4}$ μg m$^{-3}$,

respectively (Donahue et al., 2011). The saturation mass concentration of OSs spanned more than 6 orders of magnitude (Fig. 7a), suggesting their inherent chemical complexity of them. A large number of OSs are SVOCs and LVOCs while their precursors are classified as IVOCs and SVOCs, indicating that the SO$_2$ presence facilitates the reduction of product volatility (Yang et al., 2020; Han et al., 2016).

Lower-volatile OSs generated from acid-catalyzed heterogenous reactions may build





particle mass at a faster rate compared to their higher-volatile precursors, and thereby benefit the formation and growth of particles in the presence of $SO_2$.

**Figure 6.** Simplified formation schemes for the selected organosulfates formed from the ozonolysis of cyclooctene.

Figure 7b displays the DBE–carbon atom number space for organosulfur compounds. There are some overlaps of organosulfur compounds detected in this work with previous data from field observations (Wang et al., 2021; Boris et al., 2016; Cai et al., 2020). For example, Wang et al. (2021) comprehensively analyzed OS in $PM_{2.5}$ filter samples collected in an urban site in Shanghai, China and observed the presence of $C_6H_{10}O_6S$ (Fig. 7b, cyan cross). In the absence of chromatographic data such as retention times, $C_6H_{10}O_6S$ was tentatively assigned to diesel vapor-derived OS. Alkenes are important components of diesel and cyclooctene may be also responsible for $C_6H_{10}O_6S$ formation in the atmosphere. The overlaps of organosulfur compounds





indicate that the ozonolysis of cycloalkenes in the presence of $SO_2$ is likely an important source of organosulfur compounds in the ambient atmosphere. In addition, our work further suggests that the sources of OS cannot be determined only based on their

elemental formula, and techniques that enable the identification of molecular structures (e.g., MS/MS) are greatly beneficial in field studies. The identified molecular structures of OSs in this study are also helpful in the source apportionment in field studies.

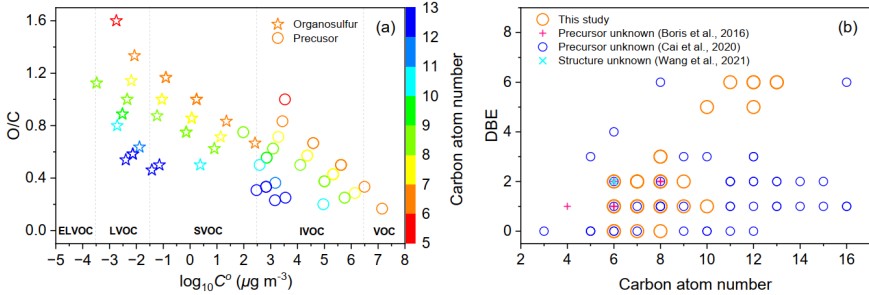

**Figure 7.** (a) Two dimensional volatility–oxidation space of the identified organosulfurs and their precursors. (b) Carbon atom number distribution of organosulfurs observed in the current work and in the studies of Cai et al. (2020), Boris et al. (2016), and Wang et al. (2021). Detailed formulae of these OSs could be found in Table S2. Organosulfurs from previous studies are of unknown origin or unknown

structure.

## 4 Conclusion

We have explored $O_3$-initiated oxidation of cyclooctene in the absence and presence of $SO_2$, with a focus on the mechanism by which $SO_2$ impacts particle formation and composition. Cyclooctene can produce a large number of particles upon

reacting with $O_3$. Higher $SO_2$ concentration led to higher particle number concentration as a result of $H_2SO_4$ formation from the reactions of sCI with $SO_2$.

Cyclooctene SOA mainly consisted of carboxylic acids and corresponding dimers with acid anhydride and aldol structures when $SO_2$ was not added. $SO_2$ addition can

induce the changes in particle chemical composition through the formation of OSs. Some OSs, classified as compounds of unknown origin or unknow structure in previous field studies, were also observed in this work. The OSs found here are less volatile than their precursors, indicating the stronger ability of OS for particle formation. The formation of OSs can in part lead to the increase in particle volume concentrations in the presence of $SO_2$.

The results here suggest that $SO_2$ can influence aerosol particle formation and composition by producing sulfur-containing compounds (i.e., $H_2SO_4$ and OSs). Nevertheless, the observed number of OSs may be amplified by the high $SO_2$ concentration used in the present work. In order to determine the actual mass yields of OSs and better quantify $SO_2$ roles in particle formation, further experiments using

ambient $SO_2$ levels and authentic standards are warranted.

**Data availability.**

Experimental data are available upon request to the corresponding author.

**Supplement.**

The supplement related to this article is available online at:

**Author contribution.**

ZY designed the experiments and carried them out. ZY performed data analysis with assistance from XL, NTT, KL, and LD. ZY prepared the paper with contributions from

all co-authors. NTT, KL, and LD commented on the paper.



**Declaration.**

The authors declare that they have no conflict of interest.

**Financial support.**

This work was supported by National Natural Science Foundation of China (no.
22076099), Youth Innovation Program of Universities in Shandong Province (no.
2019KJD007), and Fundamental Research Fund of Shandong University (no.
2020QNQT012).



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
