# Peer review of "SO2 enhances aerosol formation from anthropogenic volatile organic compound ozonolysis by producing sulfur-containing compounds"

_EGUsphere, 2022_

## Author Comment (AC1)

The authors sincerely thank the Reviewer for the careful review and thoughtful comments, which are very helpful in improving our manuscript. Comments from the Reviewer are reproduced in black font. Our point-by-point responses to the comments raised by the Reviewer are indicated in blue font below and the revised text in the manuscript is shown in red.

**Reviewer: 1**

The manuscript by Yang et al. describes a set of laboratory measurements, in which they investigated SO2 effects on the formation and chemical composition of particles from anthropogenic volatile organic compound ozonolysis. Various monomeric and dimeric products with C, H, and O atoms were observed under SO2-free conditions. The authors found that SO2 presence can induce the formation of sulfur-containing compounds. They suggested that the observed sulfur-containing compounds have stronger ability for particle formation than corresponding precursors, leading to an enhancement of particle formation. Structures and reasonable formation mechanisms of these sulfur-containing compounds were also proposed. Overall, the experimental design, results, and discussion of this manuscript are presented in a logical sequence that is easy to follow and understand. The paper provides new and valuable results for our understanding of the details of SO2 roles in aerosol formation, and also guidance and inspiration for the community that reads ACP. Therefore, I would recommend the publication of this work if the author consider the minor comments below.

Specific comments:

The authors did a great job of explaining the reasons why their study would be of importance and interest. However, there is just brief text regarding the influences of SO2 on aerosol chemistry. Some recent literatures should be considered.

Deng, P. S. J. Lakey, Y. Wang, P. Li, J. Xu, H. Pang, J. Liu, X. Xu, X. Li, X. Wang, Y. Zhang, M. Shiraiwa and S. Gligorovski, Daytime SO2 chemistry on ubiquitous urban surfaces as a source of organic sulfur compounds in ambient air, Sci. Adv., 2022, 8,

eabq6830.

Wang, T. Liu, J. Jang, J. P. D. Abbatt and A. W. H. Chan, Heterogeneous interactions between SO2 and organic peroxides in submicron aerosol, Atmos. Chem. Phys., 2021, 21, 6647-6661.

**Response:**

The literatures recommended by the Reviewer have been cited in the revised manuscript.

The following text was also added in the Introduction section to complement the influences of $SO_2$ on aerosol composition.

**Page 2, Line 39:**

Recently, the impacts of inorganic gases on aerosol chemistry have received significant attention (Deng et al., 2022).

**Page 2, Line 51:**

For instance, under humid condition, the reactive uptake of $SO_2$ onto organic aerosols was obvious and reactions of $SO_2$ with organic peroxides could contribute to organosulfate (OS) formation (Wang et al., 2021a; Ye et al., 2018). $H_2SO_4$ originated from sCI-induced oxidation of $SO_2$ is also linked to OS production (Keywood et al., 2004).

P4, L92. The authors mentioned that cyclohexane was injected into the chamber to scavenge OH radical. It is also worth mentioning how did the authors determine that OH had been successfully scavenged.

**Response:**

Aerosol yield from OH oxidation of cyclohexane is relatively low, and cyclohexane hardly reacts with $O_3$. Therefore, cyclohexane is commonly employed to isolate SOA formation from the additional impacts of OH formed in the gas-phase reaction of $O_3$ with alkenes (Bracco et al., 2019; Carlsson et al., 2012; Sato et al., 2013; Ng et al., 2006).

In this work, cyclohexane was introduced into the chamber at sufficient concentration (~130 ppm) so that more than 98% of OH generated during the ozonolysis of cyclooctene were scavenged. Control experiments were also carried out, and the corresponding results showed that the presence of cyclohexane could lead to the significant decrease in particle volume concentration (Fig. R1).

[Figure]

**Figure R1.** Particle volume concentration as a function of time during the ozonolysis of cyclooctene with and without cyclohexane addition.

For clarification, the following text was inserted in the experimental method section, and Fig. R1 was added in the revised supplement.

**Page 4, Line 95:**

Cyclohexane (99.5%, Aladdin) was injected in excess (~130 ppm) into the chamber so that more than 98% of OH generated during the ozonolysis of cyclooctene were scavenged. Control experiments showed that the presence of cyclohexane could lead to the significant decrease in particle volume concentration (Fig. S1).

P5, L118 and L128. Particle production experiments were carried out as batch mode experiments. Therefore, the chemical reaction systems were evolving during the ozonolysis of anthropogenic volatile compound. During which reaction period of the experiments did the authors collect aerosol particles? Please indicate this information.

**Response:**

Aerosol particles were collected 300 min after the reaction initiation. For clarification, the following sentence was added in the method section.

**Page 5, Line 115:**

The particle formation experiments proceeded for 300 min before the collection of aerosol particles.

P6, L130. It should be also stated clear how the extraction was done (i.e., whole filter or punches? device?)

**Response:**

The original sentence has been revised to show more details about the extraction method.

**Page 6, Line 139:**

The whole sample filters were extracted twice into 5 mL of methanol (Optima$^{®}$ LC-MS grade, Fisher Scientific) by ice sonication (KQ5200E, Kunshan Ultrasonic Instruments, China) for 20 min.

P7, L162–167. Perhaps it is better for the understanding of readers to include some relevant citations.

**Response:**

The KMD method has been widely applied for the analysis of mass spectra of organic aerosols. For clarification, the following sentence and references were added in the revised manuscript.

**Page 7, Line 178:**

The KMD value is same for homologous species that differ from each other only by their base units.

Kenseth, C. M., Hafeman, N. J., Huang, Y., Dalleska, N. F., Stoltz, B. M., and Seinfeld, J. H.: Synthesis of Carboxylic Acid and Dimer Ester Surrogates to Constrain the Abundance and Distribution of Molecular Products in alpha-Pinene and beta-Pinene Secondary Organic Aerosol, Environ. Sci. Technol., 54, 12829-12839, 10.1021/acs.est.0c01566, 2020.

Kundu, S., Fisseha, R., Putman, A. L., Rahn, T. A., and Mazzoleni, L. R.: Molecular

formula composition of β-caryophyllene ozonolysis SOA formed in humid and dry conditions, Atmos. Environ., 154, 70-81, 10.1016/j.atmosenv.2016.12.031, 2017.

P7, L168. Full equation from Li et al. (2016) should be given.

**Response:**

The following formulation has been updated in the revised manuscript.

$$\log_{10} C_i^{\mathrm{o}} = (n_{\mathrm{C}}^0 - n_{\mathrm{C}}^i)b_{\mathrm{C}} - n_{\mathrm{O}}^i b_{\mathrm{O}} - 2\frac{n_{\mathrm{C}}^i n_{\mathrm{O}}^i}{n_{\mathrm{C}}^i + n_{\mathrm{O}}^i} b_{\mathrm{CO}} - n_{\mathrm{N}}^i b_{\mathrm{N}} - n_{\mathrm{S}}^i b_{\mathrm{S}}$$

P9, L225. From Fig. S2, the volume concentration of aerosol particles reached its maximum within 240 min. So, what is the definition of the initial stage of particle production experiment?

**Response:**

Once $O_3$ was introduced into the reactor, cyclooctene began to be oxidized and aerosol particles were produced rapidly. Based on the initial concentrations of $O_3$ (~800 ppb) and cyclooctene (~200 ppb) as well as the reaction rate constant ($4.51 \times 10^{-16}$ cm$^3$ molecule$^{-1}$ s$^{-1}$), the reaction of cyclooctene with $O_3$ could be completed within several minutes. GC-FID measurements also indicated that cyclooctene was completely consumed within 10 minutes after the ozonolysis was initiated. When cyclooctene was depleted, aerosol particle mass increased slowly. Therefore, the initial stage of particle formation in this work was defined as the time (i.e., 10 min) from reaction initiation to the complete consumption of cyclooctene.

Now, the definition of the initial stage of particle production experiment is provided in the revised manuscript.

**Page 10, line 242:**

Once $O_3$ was introduced into the reactor, aerosol particles were produced rapidly. After cyclooctene was depleted, the aerosol particle mass increased slowly. The initial stage of particle formation was then defined as the time from reaction initiation to the complete consumption of cyclooctene (~10 min).

The authors mentioned that the enhancement of aerosol particles was mainly due to the formation of inorganic and organic sulfates. Although wall losses of organic vapors may be negligible as the author discussed, previous studies suggest that increased particle surface area by SO2 may cause the increase in particle volume concentration. Would this be a possible clarification?

**Response:**

As suggested by the Reviewer, condensation of organic vapors onto aerosol particles can compete with wall losses of organic vapors. To better investigate the roles of $SO_2$ via increasing particle surface in promoting particle formation, we calculated the timescale associated with gas-particle partitioning equilibrium ($\tau_{g-p}$, s) and gas-wall equilibrium ($\tau_{g-w}$, s) based on the method of Zhang et al. (2014). More details about this method are shown below.

**(a) Gas-wall equilibrium.** The wall-loss process of gas-phase products is generally taken into account to be first-order and the first-order wall-loss coefficient of gas-phase products, $k_w$ ($s^{-1}$), can be calculated as

$$k_w = \frac{A}{V} \times \frac{0.25\alpha_w\bar{c}}{1.0 + \frac{\pi}{2} \times \frac{\alpha_w\bar{c}}{4(k_eD_{gas})^{0.5}}} \tag{R1}$$

where $A/V$ (5.55 $m^{-1}$) is the surface to volume ratio of Teflon reactor; $\alpha_w$ is the mass accommodation coefficient of gas-phase products onto the inner wall and an $\alpha_w$ value of $10^{-5}$ was employed (Matsunaga and Ziemann, 2010; Zhang et al., 2014); $\bar{c}$ (m $s^{-1}$) is the molecules' mean thermal speed; $k_e$ ($s^{-1}$) and $D_{gas}$ ($m^2$ $s^{-1}$) are the eddy diffusion coefficient and the molecular diffusivity, respectively. A $k_e$ value of 0.02 $s^{-1}$ was adopted (Mcmurry and Grosjean, 1985). $D_{gas}$ was estimated to be $6 \times 10^{-6}$ $m^2$ $s^{-1}$ (Krechmer et al., 2017; Tang et al., 2015). MW (g $mol^{-1}$) is molecular weight of the given gas-phase product. An average molecular weight of 200 g $mol^{-1}$ for gas-phase products was used to estimate the influence of vapor wall loss (Sarrafzadeh et al., 2016).

The mean thermal speed could be determined according to the following equation:

$$\bar{c} = \sqrt{\frac{8RT}{\pi MW}} \tag{R2}$$

where R (8.314 J mol$^{-1}$ K$^{-1}$) and T (K) are the ideal gas constant and experimental temperature, respectively.

The time required to approach gas-wall equilibrium ($\tau_{\text{g-w}}$, s) can be calculated as:

$$\tau_{\text{g-w}} = \frac{1}{k_w} \tag{R3}$$

**(b) Gas-particle equilibrium.** The time associated with approaching gas-particle equilibrium ($\tau_{\text{g-p}}$, s) can be determined using the following equation:

$$\tau_{\text{g-p}} = \frac{1}{2\pi N_p \overline{D_p}\, D_{gas} \overline{F_{FS}}} \tag{R4}$$

where $N_p$ (# m$^{-3}$) and $\overline{D_p}$ (m) are the number concentration and mean diameter of aerosol particles, respectively; $D_{gas}$ (m$^2$ s$^{-1}$) is the molecular diffusivity. $\overline{F_{FS}}$ is the Fuchs-Sutugin correction and it is equal to:

$$\overline{F_{FS}} = \frac{0.75\alpha_p(1+k_n)}{k_n^2 + k_n + 0.283 k_n \alpha_p + 0.75\alpha_p} \tag{R5}$$

where $\alpha_p$ is the mass accommodation coefficient of gas-phase products onto aerosol particles. An $\alpha_p$ value of 0.7 was adopted (Krechmer et al., 2017). K$_n$ is the Knudsen number, which can be calculated as:

$$k_n = \frac{2\lambda}{D_p} \tag{R6}$$

The gas mean free path ($\lambda$, nm) of gas-phase product is defined as:

$$\lambda = \frac{3D_{gas}}{\overline{c}} \tag{R7}$$

The value of $\tau_{\text{g-w}}$ was determined to be around 20.4 ± 0.01 min. The estimated $\tau_{\text{g-p}}$ value decreased from 0.13 ± 0.01 to 0.07 ± 0.01 min when SO$_2$ concentrations increased from 0 to 129 ppb. Gas-particle partitioning could dominate the wall deposition of gas-phase products for particle number concentrations in our chamber experiments. Therefore, the roles of SO$_2$ via increasing particle surface in promoting particle formation may be negligible.

For clarification, the following details about the calculation of gas-particle partitioning equilibrium and gas-wall equilibrium were inserted into the revised

supplement.

**S1. Estimation of wall losses of organic vapors**

**(a) Gas-wall equilibrium.** The wall-loss process of gas-phase products is generally taken into account to be first-order and the first-order wall-loss coefficient of gas-phase products, $k_w$ (s$^{-1}$), can be calculated as

$$k_w = \frac{A}{V} \times \frac{0.25\alpha_w\bar{c}}{1.0 + \frac{\pi}{2} \times \frac{\alpha_w\bar{c}}{4(k_e D_{gas})^{0.5}}} \tag{S1}$$

where $A/V$ (5.55 m$^{-1}$) is the surface to volume ratio of Teflon reactor; $\alpha_w$ is the mass accommodation coefficient of gas-phase products onto the inner wall and an $\alpha_w$ value of 10$^{-5}$ was employed (Matsunaga and Ziemann, 2010; Zhang et al., 2014); $\bar{c}$ (m s$^{-1}$) is the molecules' mean thermal speed; $k_e$ (s$^{-1}$) and $D_{gas}$ (m$^2$ s$^{-1}$) are the eddy diffusion coefficient and the molecular diffusivity, respectively. A $k_e$ value of 0.02 s$^{-1}$ was adopted (Mcmurry and Grosjean, 1985). $D_{gas}$ was estimated to be $6 \times 10^{-6}$ m$^2$ s$^{-1}$ (Krechmer et al., 2017; Tang et al., 2015). MW (g mol$^{-1}$) is molecular weight of the given gas-phase product. An average molecular weight of 200 g mol$^{-1}$ for gas-phase products was used to estimate the influence of vapor wall loss (Sarrafzadeh et al., 2016).

The mean thermal speed could be determined according to the following equation:

$$\bar{c} = \sqrt{\frac{8RT}{\pi MW}} \tag{S2}$$

where R (8.314 J mol$^{-1}$ K$^{-1}$) and T (K) are the ideal gas constant and experimental temperature, respectively.

The time required to approach gas-wall equilibrium ($\tau_{g\text{-}w}$, s) can be calculated as:

$$\tau_{g\text{-}w} = \frac{1}{k_w} \tag{S3}$$

**(b) Gas-particle equilibrium.** The time associated with approaching gas-particle equilibrium ($\tau_{g\text{-}p}$, s) can be determined using the following equation:

$$\tau_{g\text{-}p} = \frac{1}{2\pi N_p \overline{D_p} D_{gas} \overline{F}_{FS}} \tag{S4}$$

where $N_p$ (# m$^{-3}$) and $\overline{D_p}$ (m) are the number concentration and mean diameter of aerosol particles, respectively; $D_{gas}$ (m$^2$ s$^{-1}$) is the molecular diffusivity. $\overline{F_{FS}}$ is the Fuchs-Sutugin correction and it is equal to:

$$\overline{F_{FS}} = \frac{0.75\alpha_p(1+k_n)}{k_n^2 + k_n + 0.283k_n\alpha_p + 0.75\alpha_p} \tag{S5}$$

where $\alpha_p$ is the mass accommodation coefficient of gas-phase products onto aerosol particles. An $\alpha_p$ value of 0.7 was adopted (Krechmer et al., 2017). $K_n$ is the Knudsen number, which can be calculated as:

$$k_n = \frac{2\lambda}{D_p} \tag{S6}$$

The gas mean free path ($\lambda$, nm) of gas-phase product is defined as:

$$\lambda = \frac{3D_{gas}}{\overline{c}} \tag{S7}$$

The value of $\tau_{\text{g-w}}$ was determined to be around $20.4 \pm 0.01$ min. The estimated $\tau_{\text{g-p}}$ value decreased from $0.13 \pm 0.01$ to $0.07 \pm 0.01$ min when SO$_2$ concentrations increased from 0 to 129 ppb. Gas-particle partitioning could dominate the wall deposition of gas-phase products for particle number concentrations in our chamber experiments.

P17, L378. Could you provide some more details about the IR absorption of different functional groups? Perhaps in the Supplement?

**Response:**

The IR absorption of assigned functional groups are summarized in the following Table, and they have been inserted in the revised supplement.

**Table R1.** IR absorption of functional groups.

| Assignment | Wavenumber (cm$^{-1}$) | References |
|---|---|---|
| H-bonding of OH in alcohol | 3600–3200 | (Hung et al., 2013) |
| H-bonding of carboxylic acid | 3200–2400 | (Sax et al., 2005) |
| aliphatic CH | 3000–2800 | (Sax et al., 2005) |
| C=O in carboxylic acid/ketone/aldehyde/ester | 1750–1685 | (Hung et al., 2013) |
| OH of alcohol in-plane deformation vibration | 1440–1260 | |
| C-O stretching in primary alcohol | 1090–1000 | |
| C-O stretching in secondary alcohol | 1150–1075 | |
| C-O stretching in tertiary alcohol | 1210–1100 | |
| C-O stretching in carboxylic acid | 1320–1210 | |
| C-O stretching in peroxide | 1150–1030 | |
| asymmetric SO$_2$ stretching | 1415–1370 | (Tammer, 2004) |
| C-O-C stretching | 1050–1010 | (Liu et al., 2015) |
| C-O-C stretching in acetal | 1085 | (Lal et al., 2012) |
| symmetric SO$_2$ stretching | 1064 | (Lal et al., 2012) |
| C-O vibration of C-O-S | 1050–1030 | (Hung et al., 2013) |
| asymmetric SO stretching | 1020–850 | (Hung et al., 2013) |
| C-O-C stretching in ether | 950 | (Lin et al., 2014) |
| O-O stretching in peroxide | 900–800 | (Hung et al., 2013) |
| asymmetric C-O-S stretching | 875 | (Tammer, 2004) |
| symmetric C-O-S stretching | 750 | |

P19, L 396. The Reviewer would recommend the author to present briefly the strengths of ESI-MS in characterizing organosulfate. This may be significant in supporting the production of organosulfate.

**Response:**

   Traditional analytical methods, for example GC-MS techniques, failed to measure organosulfates (Surratt et al., 2007). In contrast, UHPLC-HRMS coupled with ESI has been recognized as a robust and effective method for the analysis of organosulfate in aerosol samples (Riva et al., 2016; Bruggemann et al., 2020; Wang et al., 2021b). Organosulfates could undergo highly efficient ionization to give deprotonated

molecular ions in negative ion mode without derivatization. Orbitrap MS with high sensitivity and molecular specificity allows us to accurately assign an amount of mass spectral signals to organosulfates (Eliuk and Makarov, 2015; Bruggemann et al., 2020). Furthermore, unambiguous identification of organosulfates can be achieved based on tandem mass spectrometry analysis because organosulfates could give characteristic fragment ions at $m/z$ 80 ($SO_3^-$), 81 ($HSO_3^-$), and/or 97 ($HSO_4^-$).

The following text have been added to enrich the description about ESI-MS in the revised manuscript.

**Page 20, Line 415:**

OS could undergo highly efficient ionization to give deprotonated molecular ions in negative ion mode. Based on MS/MS analysis, unambiguous identification of OS can be achieved since OSs could give characteristic fragment ions at $m/z$ 80 ($SO_3^-$), 81 ($HSO_3^-$), and/or 97 ($HSO_4^-$) in their MS/MS spectra.

P22, Figure 7. The legend "Precursor" is confusing since cyclooctene is also referred to as a precursor in this manuscript. Suggest different legend such as organosulfate precursor or something else.

**Response:**

Legend has been updated in the revised manuscript as follows:

[Figure]

Figure 7. (a) Two dimensional volatility–oxidation space of the identified organosulfurs and their precursors. (b) Carbon atom number distribution of organosulfurs observed in the current work and in the studies of Cai et al. (2020), Boris et al. (2016), and Wang et al. (2021). Detailed formulae of these OSs can be found in Table S3. Organosulfurs from previous studies are of unknown origin or unknown structure.

**References**

Bracco, L. L. B., Tucceri, M. E., Escalona, A., Diaz-de-Mera, Y., Aranda, A., Rodriguez, A. M., and Rodriguez, D.: New particle formation from the reactions of ozone with indene and styrene, Phys. Chem. Chem. Phys., 21, 11214-11225, 10.1039/c9cp00912d, 2019.

Bruggemann, M., Xu, R., Tilgner, A., Kwong, K. C., Mutzel, A., Poon, H. Y., Otto, T., Schaefer, T., Poulain, L., Chan, M. N., and Herrmann, H.: Organosulfates in Ambient Aerosol: State of Knowledge and Future Research Directions on Formation, Abundance, Fate, and Importance, Environ. Sci. Technol., 54, 3767-3782, 10.1021/acs.est.9b06751, 2020.

Carlsson, P. T., Keunecke, C., Kruger, B. C., Maass, M. C., and Zeuch, T.: Sulfur dioxide oxidation induced mechanistic branching and particle formation during the ozonolysis of β-pinene and 2-butene, Phys. Chem. Chem. Phys., 14, 15637-15640, 10.1039/c2cp42992f, 2012.

Deng, H., Lakey, P. S. J., Wang, Y., Li, P., Xu, J., Pang, H., Liu, J., Xu, X., Li, X., Wang, X., Zhang, Y., Shiraiwa, M., and Gligorovski, S.: Daytime $SO_2$ chemistry on ubiquitous urban surfaces as a source of organic sulfur compounds in ambient air, Sci. Adv., 8, eabq6830, doi:10.1126/sciadv.abq6830, 2022.

Eliuk, S. and Makarov, A.: Evolution of Orbitrap Mass Spectrometry Instrumentation, Annu Rev Anal Chem (Palo Alto Calif), 8, 61-80, 10.1146/annurev-anchem-071114-040325, 2015.

Hung, H. M., Chen, Y. Q., and Martin, S. T.: Reactive aging of films of secondary organic material studied by infrared spectroscopy, J. Phys. Chem. A, 117, 108-116, 10.1021/jp309470z, 2013.

Keywood, M. D., Varutbangkul, V., Bahreini, R., Flagan, R. C., and Seinfeld, J. H.: Secondary organic aerosol formation from the ozonolysis of cycloalkenes and related compounds, Environ. Sci. Technol., 38, 4157-4164, 10.1021/es035363o, 2004.

Krechmer, J. E., Day, D. A., Ziemann, P. J., and Jimenez, J. L.: Direct Measurements of Gas/Particle Partitioning and Mass Accommodation Coefficients in Environmental Chambers, Environ. Sci. Technol., 51, 11867-11875, 10.1021/acs.est.7b02144, 2017.

Lal, V., Khalizov, A. F., Lin, Y., Galvan, M. D., Connell, B. T., and Zhang, R.: Heterogeneous reactions of epoxides in acidic media, J. Phys. Chem. A, 116, 6078-6090, 10.1021/jp2112704, 2012.

Lin, Y. H., Budisulistiorini, H., Chu, K., Siejack, R. A., Zhang, H. F., Riva, M., Zhang, Z. F., Gold, A., Kautzman, K. E., and Surratt, J. D.: Light-Absorbing Oligomer Formation in Secondary Organic Aerosol from Reactive Uptake of Isoprene Epoxydiols, Environ. Sci. Technol., 48, 12012-12021, 10.1021/es503142b, 2014.

Liu, Y., Liggio, J., Staebler, R., and Li, S. M.: Reactive uptake of ammonia to secondary organic aerosols: kinetics of organonitrogen formation, Atmos. Chem. Phys., 15, 13569-13584, 10.5194/acp-15-13569-2015, 2015.

Matsunaga, A. and Ziemann, P. J.: Gas-Wall Partitioning of Organic Compounds in a Teflon Film Chamber and Potential Effects on Reaction Product and Aerosol Yield Measurements, Aerosol Sci. Technol., 44, 881-892, 10.1080/02786826.2010.501044, 2010.

McMurry, P. H. and Grosjean, D.: Gas and aerosol wall losses in Teflon film smog chambers, Environ. Sci. Technol., 19, 1176-1182, 10.1021/es00142a006, 1985.

Ng, N. L., Kroll, J. H., Keywood, M. D., Bahreini, R., Varutbangkul, V., Flagan, R. C., Seinfeld, J. H., Lee, A., and Goldstein, A. H.: Contribution of first- versus second-generation products to secondary organic aerosols formed in the oxidation of biogenic hydrocarbons, Environ. Sci. Technol., 40, 2283-2297, 10.1021/es052269u, 2006.

Riva, M., Barbosa, T. D. S., Lin, Y.-H., Stone, E. A., Gold, A., and Surratt, J. D.: Chemical characterization of organosulfates in secondary organic aerosol derived from the photooxidation of alkanes, Atmos. Chem. Phys., 16, 11001-11018, 10.5194/acp-16-11001-2016, 2016.

Sarrafzadeh, M., Wildt, J., Pullinen, I., Springer, M., Kleist, E., Tillmann, R., Schmitt, S. H., Wu, C., Mentel, T. F., Zhao, D., Hastie, D. R., and Kiendler-Scharr, A.: Impact of $NO_x$ and OH on secondary organic aerosol formation from β-pinene photooxidation, Atmos. Chem. Phys., 16, 11237-11248, 10.5194/acp-16-11237-2016, 2016.

Sato, K., Inomata, S., Xing, J.-H., Imamura, T., Uchida, R., Fukuda, S., Nakagawa, K., Hirokawa, J., Okumura, M., and Tohno, S.: Effect of OH radical scavengers on secondary organic aerosol formation from reactions of isoprene with ozone, Atmos. Environ., 79, 147-154, 10.1016/j.atmosenv.2013.06.036, 2013.

Sax, M., Zenobi, R., Baltensperger, U., and Kalberer, M.: Time resolved infrared spectroscopic analysis of aerosol formed by photo-oxidation of 1,3,5-trimethylbenzene and alpha-pinene, Aerosol Sci. Technol., 39, 822-830, 10.1080/02786820500257859, 2005.

Surratt, J. D., Kroll, J. H., Kleindienst, T. E., Edney, E. O., Claeys, M., Sorooshian, A., Ng, N. L., Offenberg, J. H., Lewandowski, M., Jaoui, M., Flagan, R. C., and Seinfeld, J. H.: Evidence for organosulfates in secondary organic aerosol, Environ. Sci. Technol., 41, 517-527, 10.1021/es062081q, 2007.

Tammer, M.: G. Sokrates: Infrared and Raman characteristic group frequencies: tables and charts, Colloid and Polymer Science, 283, 235-235, 10.1007/s00396-004-1164-6, 2004.

Tang, M. J., Shiraiwa, M., Pöschl, U., Cox, R. A., and Kalberer, M.: Compilation and evaluation of gas phase diffusion coefficients of reactive trace gases in the atmosphere: Volume 2. Diffusivities of organic compounds, pressure-normalised mean free paths, and average Knudsen numbers for gas uptake calculations, Atmos. Chem. Phys., 15, 5585-5598, 10.5194/acp-15-5585-2015, 2015.

Wang, S., Liu, T., Jang, J., Abbatt, J. P. D., and Chan, A. W. H.: Heterogeneous interactions between SO2 and organic peroxides in submicron aerosol, Atmos. Chem. Phys., 21, 6647-6661, 10.5194/acp-21-6647-2021, 2021a.

Wang, Y., Zhao, Y., Wang, Y., Yu, J.-Z., Shao, J., Liu, P., Zhu, W., Cheng, Z., Li, Z., Yan, N., and Xiao, H.: Organosulfates in atmospheric aerosols in Shanghai, China: seasonal and interannual variability, origin, and formation mechanisms, Atmos. Chem. Phys., 21, 2959-2980, 10.5194/acp-21-2959-2021, 2021b.

Ye, J., Abbatt, J. P. D., and Chan, A. W. H.: Novel pathway of SO2 oxidation in the atmosphere: reactions with monoterpene ozonolysis intermediates and secondary organic aerosol, Atmos. Chem. Phys., 18, 5549-5565, 10.5194/acp-18-5549-2018, 2018.

Zhang, X., Cappa, C. D., Jathar, S. H., McVay, R. C., Ensber, J. J., Kleeman, M. J., and Seinfeld, J. H.: Influence of vapor wall loss in laboratory chambers on yields of secondary organic aerosol, P. Natl. Acad. Sci. USA, 111, 5802-5807, 2014.

---

## Author Comment (AC2)

The authors sincerely thank the Reviewer for the careful review and thoughtful comments, which are very helpful in improving our manuscript. Comments from the Reviewer are reproduced in black font. Our point-by-point responses to the comments raised by the Reviewer are indicated in blue font below and the revised text in the manuscript is shown in red.

**Reviewer: 2**

General comments:

This paper describes the enhancement of aerosol formation in the presence of SO2 during cyclooctene ozonolysis. The composition of the formed SOA was investigated by means of ATR-FTIR and LC-MS/MS and the authors found that the enhancement was largely attributed to the formation of H2SO4 and organosulfates (OSs). By using high-resolution MS/MS, the molecular structures of many OSs were proposed in this work. I think that this study was well-conducted and that the data presented here are valuable for the understanding of the SOA formation. In addition, the paper is generally well-written. I recommend this paper to be published in Atmospheric Chemistry and Physics after the authors' consideration of my minor comments detailed below.

Specific comments:

Page 4, Section 2.1: It is better to show the rate constant for the reaction of cyclooctene with O3.

**Response:**

We provided the reaction rate constant ($k$) for the reaction of cyclooctene with $O_3$ ($k_{298\ K} = 4.51 \times 10^{-16}$ $cm^3$ $molecule^{-1}$ $s^{-1}$) in the Introduction section. For clarity, this rate constant was also showed in Section 2.1 in the revised manuscript.

Page 5, Table 1: There is no information about the reaction time. Is it 300 min?

**Response:**

Particle formation experiments were operated in a batch mode and aerosol particles were collected at 300 min after reaction initiation. For clarification, the following sentence was added in the method section.

**Page 5, Line 115:**

The particle formation experiments proceeded for 300 min before the collection of aerosol particles.

I guess that the reaction of cyclooctene with O3 was completed within several minutes. Why did the authors measure for such the long reaction time?

**Response:**

The reaction of cyclooctene with $O_3$ is rapid. Once $O_3$ was introduced into the reactor, cyclooctene began to be oxidized and aerosol particles were produced accordingly.

Here, particle production experiments were performed in a batch mode. Therefore, the reaction systems were complicated over the course of particle formation. Cyclooctene could be completely consumed within 10 min based on GC-FID measurements but aerosol growth did not stop at the time when cyclooctene was depleted. As shown in Fig. R1, the uncorrected particle volume concentration can reach its maximum at 100 min after reaction initiation and remain constant at 100–200 min. If the apparent aerosol volume concentration measured by SMPS reaches its maximum and remains constant, it means aerosol production has not stopped yet. We would expect the aerosol volume concentration to go down once the wall loss processes take over. Thus, the aerosol volume concentration was measured continuously until we observed the decrease in aerosol volume concentration within 300 min. After wall loss correction was applied, the particle volume concentration reached its maximum at 240 min (Fig. R1).

[Figure]

**Figure R1.** Time series of the volume concentration of aerosol particles during the ozonolysis of cyclooctene.

To further clarify the goal of our experiments, the following text has been inserted in the revised manuscript

**Page 4, Line 91:**

Particle formation experiments were operated in batch mode.

**Page 4, Line 114:**

The particle volume concentration was measured continuously until we observed a decrease.

Page 17, Lines 383-387: In Hawkins et al (2010), an absorption band at 876 cm-1 was mentioned for organosulfates. There is no mention about an absorption band of organosulfates in Coury and Dillner (2008). How did the authors attribute absorption bands at 1413 and 1095 cm-1 to organosulfates? Is there any additional evidence?

**Response:**

We summarized IR absorption peak assignments in Table R1. The IR spectra of organosulfates may have different absorption bands. For example, asymmetric and symmetric stretching of $-SO_2-$ could result in strong absorbances at $1415–1370$ cm$^{-1}$ and $\sim1064$ cm$^{-1}$, respectively. The S=O vibration occurs at $1020–850$ cm$^{-1}$. The asymmetric and symmetric stretching vibration of C-O-S could contribute to weak absorbances around at $875$ cm$^{-1}$ and $750$ cm$^{-1}$, respectively. It should be noted that the

specific positions of these absorption bands could be affected by electronegative substituents and the nature of alkyl groups (Tammer, 2004).

**Table R1.** IR absorption of functional groups.

| Assignment | Wavenumber (cm⁻¹) | References |
|---|---|---|
| H-bonding of OH in alcohol | 3600–3200 | (Hung et al., 2013) |
| H-bonding of carboxylic acid | 3200–2400 | (Sax et al., 2005) |
| aliphatic CH | 3000–2800 | (Sax et al., 2005) |
| C=O in carboxylic acid/ketone/aldehyde/ester | 1750–1685 | (Hung et al., 2013) |
| OH of alcohol in-plane deformation vibration | 1440–1260 | |
| C-O stretching in primary alcohol | 1090–1000 | |
| C-O stretching in secondary alcohol | 1150–1075 | |
| C-O stretching in tertiary alcohol | 1210–1100 | |
| C-O stretching in carboxylic acid | 1320–1210 | |
| C-O stretching in peroxide | 1150–1030 | |
| asymmetric $SO_2$ stretching | 1415–1370 | (Tammer, 2004) |
| C-O-C stretching | 1050–1010 | (Liu et al., 2015) |
| C-O-C stretching in acetal | 1085 | (Lal et al., 2012) |
| symmetric $SO_2$ stretching | 1064 | (Lal et al., 2012) |
| C-O vibration of C-O-S | 1050–1030 | (Hung et al., 2013) |
| asymmetric SO stretching | 1020–850 | (Hung et al., 2013) |
| C-O-C stretching in ether | 950 | (Lin et al., 2014) |
| O-O stretching in peroxide | 900–800 | (Hung et al., 2013) |
| asymmetric C-O-S stretching | 875 | (Tammer, 2004) |
| symmetric C-O-S stretching | 750 | |

Here, the IR absorptions by sulfur-containing functional groups were detected at 1413, 1095, and 617 cm⁻¹ in the ATR-FTIR spectra of particles formed in the presence of $SO_2$. Xu et al. (2021) measured the functional groups of β-pinene SOA in the absence and presence of $SO_2$ and also observed similar IR results. Absorption bands at 1413 and 1095 may be associated with the asymmetric and symmetric stretching of -$SO_2$- while inorganic sulfates could give rise to strong absorption at 617 cm⁻¹. In the study of

Hawkins et al. (2010), the absorbance of C-O-S group at 876 cm$^{-1}$ was measured and used to quantify the relative contribution of organosulfates to organic aerosols. However, detectable absorption bands of C-O-S did not appear in our spectra. Maria et al. (2003) showed that the IR absorptivity of C-O-S group at 876 cm$^{-1}$ (0.031) is 10 times smaller than that of inorganic sulfate at 618 cm$^{-1}$ (0.41). The weak absorption peak of C-O-S at ~876 cm$^{-1}$ may be overlapping by the absorption of other functional groups and is thus difficult to identify. Therefore, we further performed high-resolution MS measurements of aerosol particles to determine whether organosulfates were produced in the presence of $SO_2$.

We have double checked our citation and the inappropriate reference that was cited accidentally has been deleted. Now, right references have been added in the revised manuscript.

Table R1 was added as Table S1 in the revised supplement, and the original text was revised as follows:

**Page 18, Line 401:**

Three additional absorption bands at 1413, 1095, and 617 cm$^{-1}$ were observed in ATR-FTIR spectra of particles formed with the introduction of $SO_2$ (Tammer, 2004; Lal et al., 2012). Absorption bands at 1413 and 1095 cm$^{-1}$ may be associated with the asymmetric and symmetric stretching of -$SO_2$- while inorganic sulfates could give rise to strong absorption at 617 cm$^{-1}$. The presence of absorption band of sulfur-containing groups suggests that $SO_2$ addition can result in the production of sulfur-containing compounds.

Page 21, Figure 6: In this figure, the formation of many kinds of compounds having hydroxy groups is proposed. Actually, many of ion signals obtained by LC-MS/MS were assigned to compounds having hydroxy groups. But it seems that the peak of alcohol-COH in the ATR-FTIR (3500-3200 cm-1) is quite smaller than that of carbonyl at 1702 cm-1. Is it reasonable?

**Response:**

FTIR has been widely employed to investigate the chemical composition of SOA. Similar ATR-FTIR spectra of SOA were observed in previous laboratory studies (Fig. R2). For example, Zhao et al. (2016) identified multifunctional compounds containing hydroxy groups as important components of SOA formed from the ozonolysis of α-cedrene (Table R2). Carbonyl group also had strongest intensity among all characterized functional groups (Fig. R2b).

[Figure]

**Figure R2.** ATR-FTIR spectra for SOA generated from ozonolysis of different alkenes. (a) γ-terpinene SOA (Xu et al., 2020); (b) α-cedrene SOA (Zhao et al., 2016); (c) α-pinene SOA (Kidd et al., 2014); (d) β-pinene SOA (Xu et al., 2021); (e) α-pinene SOA (Vander Wall et al., 2020); (f) cyclooctene SOA (this work). Initial experimental conditions are given in detail in Table R3.

**Table R2.** Potential structures of identified monomers and dimers in Zhao et al. (2016).

[Figure]

**Table R3.** Initial experimental conditions of SOA formation from the ozonolysis of alkenes.

| VOC type | [VOC]$_0$ | [O$_3$]$_0$ | T | RH | OH scavenger | Reference |
|---|---|---|---|---|---|---|
| | (ppb) | (ppb) | (K) | (%) | | |
| γ-terpinene | 302 | 1236 | 292 | 17 | cyclohexane | (Xu et al., 2020) |
| α-cedrene | 215 | 1500 | 295 ±1 | < 5 | / | (Zhao et al., 2016) |
| α-pinene | 1000 | 1000 | 297 ± 2 | < 3 | / | (Kidd et al., 2014) |
| β-pinene | 154 | 624 | 300 | 22 | cyclohexane | (Xu et al., 2021) |
| α-pinene | 250 | 250–300 | 295–298 | < 5 | cyclohexane | (Vander Wall et al., 2020) |
| cyclooctene | 195 | 839 | 296 | 25 | cyclohexane | This work |

In this work, we used FITR to determine the overall functional groups of aerosol particles rather than individual oxidized products. At the molecular level, SOA is significantly complex as it may include hundreds to thousands of oxidized products. The results shown in Fig. 6 do not represent the entire aerosol components or compounds of high concentration but are some possible precursors with alcohol group for organosulfate formation. Carbonyl groups are also present in these alcohol skeletons. Furthermore, the ozonolysis of alkene could produce organic species containing one or more carbonyl groups (Mackenzie-Rae et al., 2018; Zhao et al., 2016). Therefore, it is reasonable that the peak intensity of alcohol group is lower than that of carbonyl group.

Page 22, Figure 7: I think that the calculation of DBE (eqn. (1)) cannot be applied to organosulfates. I think that the DBE of precursors of OSs is meaningful.

**Response:**

The DBE value of a compound could reflect its unsaturation degree. For a given compound with elemental composition of $C_cH_hO_oN_nS_s$, the DBE value (number of rings and double bonds) could be calculated based on the following equation (Deng et al., 2022; Vandergrift et al., 2022; Wang et al., 2019).

$$DBE = 1 + c + \frac{n - h}{2} \tag{R1}$$

We have double checked and read our references. Many studies on organosulfate formation used eq. R1 to calculate the DBE value of organosulfates and the two S=O bonds in each sulfate group of organosulfates were not taken into account (Wang et al., 2016; Riva et al., 2016; Kuang et al., 2016). Therefore, the DBE value of organosulfates reflects the degree of unsaturation for the side carbon chain. For example, in the study of Riva et al. (2016), the $C_{10}H_{18}O_6S$ organosulfate was characterized during the photooxidation of dodecane. Riva et al. (2016) reported that $C_{10}H_{18}O_6S$ corresponded to two DBEs, which arose from a six-membered ring and an internal carbonyl group (Fig. R3).

**Figure R3.** Structure of organosulfate ($C_{10}H_{18}O_6S$), which was extracted from Figure 2 in Riva et al. (2016).

We calculated the DBE value of organosulfates for the same consideration as in the previous studies (Wang et al., 2016; Riva et al., 2016; Kuang et al., 2016). The following notes have been added in the revised manuscript.

**Page 7, Line 172:**

For organosulfate, the two S=O bonds in the sulfate group were not considered based on calculations in previous studies (Wang et al., 2016; Riva et al., 2016; Kuang et al., 2016). The DBE value of organosulfate reflects the unsaturation degree of its side carbon chain.

**References**

Deng, H., Lakey, P. S. J., Wang, Y., Li, P., Xu, J., Pang, H., Liu, J., Xu, X., Li, X., Wang, X., Zhang, Y., Shiraiwa, M., and Gligorovski, S.: Daytime $SO_2$ chemistry on ubiquitous urban surfaces as a source of organic sulfur compounds in ambient air, Sci. Adv., 8, eabq6830, doi:10.1126/sciadv.abq6830, 2022.

Hung, H. M., Chen, Y. Q., and Martin, S. T.: Reactive aging of films of secondary organic material studied by infrared spectroscopy, J. Phys. Chem. A, 117, 108-116, 10.1021/jp309470z, 2013.

Kidd, C., Perraud, V., and Finlayson-Pitts, B. J.: New insights into secondary organic aerosol from the ozonolysis of alpha-pinene from combined infrared spectroscopy and mass spectrometry measurements, Phys. Chem. Chem. Phys., 16, 22706-22716, 10.1039/c4cp03405h, 2014.

Kuang, B. Y., Lin, P., Hu, M., and Yu, J. Z.: Aerosol size distribution characteristics of organosulfates in the Pearl River Delta region, China, Atmos. Environ., 130, 23-35, 10.1016/j.atmosenv.2015.09.024, 2016.

Lal, V., Khalizov, A. F., Lin, Y., Galvan, M. D., Connell, B. T., and Zhang, R.: Heterogeneous reactions of epoxides in acidic media, J. Phys. Chem. A, 116, 6078-6090, 10.1021/jp2112704, 2012.

Lin, Y. H., Budisulistiorini, H., Chu, K., Siejack, R. A., Zhang, H. F., Riva, M., Zhang, Z. F., Gold, A., Kautzman, K. E., and Surratt, J. D.: Light-Absorbing Oligomer Formation in Secondary Organic Aerosol from Reactive Uptake of Isoprene Epoxydiols, Environ. Sci. Technol., 48, 12012-12021, 10.1021/es503142b, 2014.

Liu, Y., Liggio, J., Staebler, R., and Li, S. M.: Reactive uptake of ammonia to secondary organic aerosols: kinetics of organonitrogen formation, Atmos. Chem. Phys., 15, 13569-13584, 10.5194/acp-15-13569-2015, 2015.

Mackenzie-Rae, F. A., Wallis, H. J., Rickard, A. R., Pereira, K. L., Saunders, S. M., Wang, X., and Hamilton, J. F.: Ozonolysis of α-phellandrene – Part 2: Compositional analysis of secondary organic aerosol highlights the role of stabilised Criegee intermediates, Atmos. Chem. Phys., 18, 4673-4693, 10.5194/acp-18-4673-2018, 2018.

Maria, S. F., Russell, L. M., Turpin, B. J., Porcja, R. J., Campos, T. L., Weber, R. J., and Huebert, B. J.: Source signatures of carbon monoxide and organic functional groups in Asian Pacific Regional Aerosol Characterization Experiment (ACE-Asia) submicron aerosol types, J. Geophys. Res. Atmos., 108, 10.1029/2003jd003703, 2003.

Riva, M., Barbosa, T. D. S., Lin, Y.-H., Stone, E. A., Gold, A., and Surratt, J. D.: Chemical characterization of organosulfates in secondary organic aerosol derived from the photooxidation of alkanes, Atmos. Chem. Phys., 16, 11001-11018, 10.5194/acp-16-11001-2016, 2016.

Sax, M., Zenobi, R., Baltensperger, U., and Kalberer, M.: Time resolved infrared spectroscopic analysis of aerosol formed by photo-oxidation of 1,3,5-trimethylbenzene and alpha-pinene, Aerosol Sci. Technol., 39, 822-830, 10.1080/02786820500257859, 2005.

Tammer, M.: G. Sokrates: Infrared and Raman characteristic group frequencies: tables and charts, Colloid and Polymer Science, 283, 235-235, 10.1007/s00396-004-1164-6, 2004.

Vander Wall, A. C., Perraud, V., Wingen, L. M., and Finlayson-Pitts, B. J.: Evidence for a kinetically controlled burying mechanism for growth of high viscosity secondary organic aerosol, Environ Sci Process Impacts, 22, 66-83, 10.1039/c9em00379g, 2020.

Vandergrift, G. W., Shawon, A. S. M., Dexheimer, D. N., Zawadowicz, M. A., Mei, F., and China, S.: Molecular Characterization of Organosulfate-Dominated Aerosols over Agricultural Fields from the Southern Great Plains by High-Resolution Mass Spectrometry, ACS Earth Space Chem., 6, 1733-1741, 10.1021/acsearthspacechem.2c00043, 2022.

Wang, K., Zhang, Y., Huang, R. J., Wang, M., Ni, H., Kampf, C. J., Cheng, Y., Bilde, M., Glasius, M., and Hoffmann, T.: Molecular Characterization and Source Identification of Atmospheric Particulate Organosulfates Using Ultrahigh Resolution Mass Spectrometry, Environ. Sci. Technol., 53, 6192-6202, 10.1021/acs.est.9b02628, 2019.

Wang, X. K., Rossignol, S., Ma, Y., Yao, L., Wang, M. Y., Chen, J. M., George, C., and Wang, L.: Molecular characterization of atmospheric particulate organosulfates in three megacities at the middle and lower reaches of the Yangtze River, Atmos. Chem. Phys., 16, 2285-2298, 10.5194/acp-16-2285-2016, 2016.

Xu, L., Yang, Z., Tsona, N. T., Wang, X., George, C., and Du, L.: Anthropogenic-Biogenic Interactions at Night: Enhanced Formation of Secondary Aerosols and Particulate Nitrogen- and Sulfur-Containing Organics from beta-Pinene Oxidation, Environ. Sci. Technol., 10.1021/acs.est.0c07879, 2021.

Xu, L., Tsona, N. T., You, B., Zhang, Y., Wang, S., Yang, Z., Xue, L., and Du, L.: NOx enhances secondary organic aerosol formation from nighttime γ-terpinene ozonolysis, Atmos. Environ., 225, 117375, 10.1016/j.atmosenv.2020.117375, 2020.

Zhao, Y., Wingen, L. M., Perraud, V., and Finlayson-Pitts, B. J.: Phase, composition, and growth mechanism for secondary organic aerosol from the ozonolysis of alpha-cedrene, Atmos. Chem. Phys., 16, 3245-3264, 10.5194/acp-16-3245-2016, 2016.